# Mix-and-Match COVID-19 Vaccinations (Heterologous Boost): A Review

Ishan Garg *, Abu Baker Sheikh , Suman Pal  and Rahul Shekhar *

Department of Internal Medicine, University of New Mexico Health Sciences Center, Albuquerque, NM 87106, USA; absheikh@salud.unm.edu (A.B.S.); spal@salud.unm.edu (S.P.)
* Correspondence: ishangargmd@gmail.com (I.G.); rshekhar@salud.unm.edu (R.S.)

**Abstract:** Various safe and effective COVID-19 vaccines utilizing different platforms (mRNA, adenovirus vector, inactivated virus-based) are available against SARS-CoV-2 infection. A prime-boost regimen (administration of two doses) is recommended to induce an adequate and sustained immune response. Most of these vaccines follow a homologous regimen (the same type of vaccine as priming and booster doses). However, there is a growing interest in a heterologous prime-boost vaccination regimen to potentially help address concerns posed by fluctuating vaccine supplies, serious adverse effects (anaphylaxis and thromboembolic episodes following adenovirus-based vaccines), new emerging virulent strains, inadequate immune response in immunocompromised individuals, and waning immunity. Various studies have demonstrated that heterologous prime-boost vaccination may induce comparable or higher antibody (spike protein) titers and a similar reactogenicity profile to the homologous prime-boost regimen. Based on these considerations, the Center for Disease Control and Prevention has issued guidance supporting the "mix-and-match" heterologous boost COVID-19 vaccine strategy.

**Keywords:** COVID-19; COVID-19 vaccine; booster; heterologous prime-boost; mix-and-match





## 1. Introduction

The coronavirus disease 2019 (COVID-19) pandemic has wreaked havoc on human life and globally overwhelmed healthcare systems. It is caused by the SARS-CoV-2 virus. As of November 2021, COVID-19 has claimed the lives of 5.1 million people and infected over 255 million people worldwide [1]. Fortunately, multiple safe and effective COVID-19 vaccines are now available for public use due to the collaborative effort of the scientific community, the federal government, and pharmaceutical companies.

Several COVID-19 vaccines have been approved under the emergency use listing (EUL) by the World Health Organization (WHO), including (1) messenger RNA (mRNA): BNT162b2 Pfizer BioNTech (Pfizer, Inc.; Philadelphia, PA, USA) and mRNA-1273 Moderna vaccines (ModernaTX, Inc.; Cambridge, MA, USA); (2) viral vector (adenovirus): ChAdOx1 (ChAd, AZD1222, AstraZeneca/Oxford, UK) and Janssen Ad26.COV2.S (Janssen Biotech, Inc.; A Janssen Pharmaceutical company, Johnson & Johnson; New Brunswick, NJ, USA); and (3) inactivated virus: Sinopharm (China National Pharmaceutical Group, Beijing, China) and Sinovac (Sinovac Biotech Ltd.; Beijing, China) [2,3]. In addition, various potential vaccine candidates are under investigation. As of October 2021, there are 107 vaccines in clinical trials on humans (of which 41 have reached the final stages of testing) and at least 75 preclinical vaccines under investigation on animals [4].

Most COVID-19 vaccines require a two-dose prime-boost regimen strategy (except for the Janssen vaccine, which is currently used as a single dose) administered 3 to 12 weeks apart to provide adequate immunity to the individual. A prime-boost immunization strategy is defined as immunization with prime and booster doses of an immunogen (same immunogen = homologous, different immunogen = heterologous). Various factors, including targeted antigens (SARS-CoV-2 spike protein), delivery platforms (mRNA, viral vector,

whole virus), doses, adjuvants, route of administration, and intervals between the primary and booster dose, can impact the immunization outcome [5–7]. The primary objective of any immunization regimen is to induce an adequate, long-lasting effect against the pathogen (SARS-CoV-2 virus). Most of these approved vaccines follow a homologous prime-boost regimen (the same type of vaccine is administered as both the priming and booster dose). However, there is growing international interest in the heterologous prime-boost COVID-19 vaccine schedule to potentially help address concerns posed by fluctuating vaccine supplies, serious adverse effects (to individuals primed with ChAdOx1 vaccine), new emerging virulent strains, and waning immunity, by providing a robust immune response with a safe reactogenicity profile [8–13]. Some preliminary findings from COVID-19 vaccine studies and other non-COVID-19 vaccines suggest that a heterologous prime-boost regimen can be more immunogenic than a homologous prime-boost regimen [10,14–22].

Based on the above considerations, the purpose of this article is to provide an in-depth review of the heterologous prime-boost COVID-19 vaccine regimen. The immunogenicity and reactogenicity of a heterologous prime-boost COVID-19 vaccine regimen, compared to the existing homologous regimen, is discussed.

## 2. Heterologous Prime-Boost Regimen: Immunogenicity and Reactogenicity

A heterologous prime-boost strategy is not a new concept. It is used as an effective approach to increase immune responses to a wide variety of infectious diseases, including human immunodeficiency virus (HIV), hepatitis C virus (HCV), hepatitis B virus (HBV), and herpes simplex virus (HSV). These immunization protocols combine the induction of antibody and CD4+ T-cell responses by proteins, stimulation of cellular immunity by DNA vaccines, enhancement of CD4+ and CD8+ T-cells, and antibodies by recombinant viral vectors [5,6,23–25]. Some of these prime-boost combinations have progressed to clinical trials with promising results. For instance, in studies conducted on vaccines against the Ebola virus, heterologous vaccination produced a robust and sustained elevation of specific immunity without producing any vaccine-related serious adverse events [26–28]. Importantly, in many vaccines, the heterologous prime-boost is more immunogenic than the homologous prime-boost regimen, although the exact mechanism behind this is not well understood [14–16,29,30]. Some studies have suggested that priming with a DNA-based vaccine may improve the avidity of the antibody response to a protein-based vaccine [31,32]. It is hypothesized that in vivo production of antigens by DNA or mRNA-based vaccines may lead to the production of memory B cells specific for the conformational domains of the antigen [5]. Supportive evidence was reported by Vaine et al. in a rabbit model [33]. They demonstrated the presence of conformational sensitive antibodies following priming by a DNA-based vaccine (for in vivo delivery of HIV-1 gp120 antigen).

Different platforms have been used to produce COVID-19 vaccines, including nucleic acid (DNA and RNA), viral vectors, protein subunits, and whole-virus platforms. For instance, the mRNA vaccines (BNT162b2 Pfizer BioNTech, mRNA-1273 Moderna) encode for the SARS-CoV-2 spike protein, viral vector vaccines (ChAdOx1 AstraZeneca/Oxford, Janssen/Ad26.COV 2.S) use adenovirus-based vectors, and inactivated whole-virus vaccine (Sinopharm, Sinovac-CoronaVac) use the whole virus to deliver genetic material (encoding for SARS-CoV-2 spike protein) to host cells. The host cells make copies of the coronavirus protein (spike protein) to generate an immune response, producing T-lymphocytes and antibodies against the viral antigen (spike protein) [13]. Findings from selected studies are summarized in Table 1.

**Table 1.** Summary of included studies on heterologous COVID-19 vaccination.

| Study | Regimen (Prime-Boost) | Country | Design | Participants (N) | Age (Years) | Common Side Effects (%) | Severe Adverse Events | Interval Boost Duration | Cellular Immune Response | Humoral Immune Response | Outcome | Funding | Declaration of Interests |
|---|---|---|---|---|---|---|---|---|---|---|---|---|---|
| Logunov et al. (NCT04436471, NCT04437875) [34] | Sputnik V (rAd26 and rAd5) | Russia | Open, non-randomised phase 1/2 | 76 healthy adults(23 women and 53 men) | 18–60 | Pain at injection site (58%), hyperthermia (50%), headache (42%), asthenia (28%), muscle and joint pain (24%) | None | 21 days | All participants, with median cell proliferation of 2.5% CD4+ and 1.3% CD8+ | All participants produced antibodies to SARS-CoV-2 glycoprotein (100% sero-conversion) | Good safety profile and induced strong humoral–cellular immune responses | Ministry of Health of the Russian Federation | Yes |
| Logunov et al. (NCT04530396) [35] | Sputnik V | Russia (Multicenter) | Randomised, double-blind, placebo-controlled, phase 3 trial | 21,977 healthy adults (vaccine group: 14,964 **—5821 women and 9143 men; placebo group: 4902 **—1887 women and 3015 men) | ≥18 | Flu-like illness, injection site reactions, headache, and asthenia | 68 participants (45- vaccine group, 23 placebo group) * | 21 days | All participants in the vaccine group had significantly higher levels of IFN-γ | 98·25% sero-conversion (four-fold increase in titre at 42 days compared with the day before first vaccination) in the vaccine group (participants produced antibodies to SARS-CoV-2 glycoprotein) | Good safety profile and induced strong humoral–cellular immune responses.May also induce broad antibody response with ability to recognize wide variety of epitopes on SARS-CoV-2 glycoprotein S | Moscow City Health Department, Russian Direct Investment Fund, and Sberbank | Yes |

**Table 1.** *Cont.*

| Study | Regimen (Prime-Boost) | Country | Design | Participants (N) | Age (Years) | Common Side Effects (%) | Severe Adverse Events | Interval Boost Duration | Cellular Immune Response | Humoral Immune Response | Outcome | Funding | Declaration of Interests |
|---|---|---|---|---|---|---|---|---|---|---|---|---|---|
| Liu et al. (ISRCTN, 69254139) [12] | ChAd and BNT | UK (Multicenter) | Randomised, non-inferiority trial | 830 (adults with no or well controlled co-morbidities) (463 prime-boost regimen—212 women, and 251 men) | >50 | Both heterologous schedules (at 28- and 84-day prime-boost intervals) induced greater systemic and local reactogenicity than their homologous counterparts | 4 * | 28-day (463 participants) and 84-day (367 participants) | NA | NA | These data support flexibility in use of heterologous prime-boost vaccination (using ChAd and BNT COVID-19 vaccines) | UK Vaccine Task Force and National Institute for Health Research | Yes |
| Borobia et al. (NCT04860739) [10] | ChAd (prime) and BNT (booster) | Spain (Multicenter) | Phase 2, open label, randomised, controlled trial | 676 (382 women and 294 men) | 18–60 | Injection site pain (88%), induration (35%), headache (44%), and myalgia (43%) | None | 8–12 weeks (61%: 8–9 weeks; 39%: 10–12 weeks) | All participants in the vaccine group had significantly higher levels of IFN-γ | All participants produced in the vaccine group produced antibodies to SARS-CoV-2 glycoprotein (100% sero-conversion; >0.8 BAU/mL) | Induced strong humoral–cellular immune responses, with an acceptable and manageable reactogenicity profile | Instituto de Salud Carlos III | Yes |
| Atmar et al. (NCT04889209) [36] | mRNA-1273, ChAd, and BNT | US (Multicenter) | Open, non-randomised phase 1/2 | 458 healthy adults (231 women and 227 men) (Booster-154 mRNA-1273, 150 ChAd, and 153 BNT) | ≥18 | Site pain, malaise, headache, and myalgiaSimilar reactogenicity profile between the groups | NA | 21–28 days | Stimulated an anamnestic response in persons who previously received primary series of any of these vaccines | Heterologous regimens increased antibody titers by 6.2–76-times, compared to the 4.2–20-times increase after homologous regimen | | NIH, CIVICs | None |

rAd = recombinant adenoviruses vectors; CD = cluster of differentiation; SARS-CoV-2 = severe acute respiratory syndrome coronavirus 2; BNT = BNT162b2 vaccine, Pfizer–BioNTech; ChAd = ChAdOx1 nCoV-19 vaccine, AstraZeneca; IFN-γ = interferon gamma; BAU = binding antibody units; * = serious adverse events were considered not to be related to vaccination; ** = participants included in primary outcome analysis; NIH = National Institute of Health; NIAID = National Institute for Allergy and Infectious Diseases; CIVICs = NIAID Collaborative Influenza Vaccine Innovation Centers.

Initial interest in heterologous COVID-19 vaccine strategy rose due to a severe side-effect, thrombotic events with thrombocytopenia, in people vaccinated with ChAdOx1-S (AstraZeneca) and Ad26.COV2.S (Janssen) vaccines [37–39]. Due to this life-threatening side-effect, health authorities were forced to modify immunization guidelines, including administration of the heterologous vaccine booster BNT162b2 (Pfizer BioNTech) in people who had received ChAdOx1-S as the primary dose [22]. Data from animal studies on COVID-19 vaccine heterologous prime-boost have shown robust immune response with increased SARS-CoV-2 IgG specific titers with neutralization ability and a robust T-helper-1-type response on either ChAdOx1-S or BNT162b2 as prime or booster doses [22,40]. Human data on COVID-19 heterologous prime-boost are trickling in following the adoption of heterologous regimens in the US and several European countries.

### 2.1. Sputnik V

Gam-COVID-Vac (Sputnik V), developed by Gamaleya National Research Centre for Epidemiology and Microbiology, Moscow, Russia, has adopted a unique strategy, using two different recombinant adenovirus (rAd) vectors—type 26 (rAd26) and type 5 (rAd5)—both carrying the gene for SARS-CoV-2 spike glycoprotein (rAd26-S and rAd5-S). The vaccine is administered (0·5 mL/dose) intramuscularly in a prime-boost regimen: a 21-day interval between the first dose (rAd26) and the second dose (rAd5). The goal of the strategy was to produce a durable and long-lasting immune response using a heterologous prime-boost vaccination strategy [5,41–43].

The initial results from phase $\frac{1}{2}$ clinical trials showed that the vaccine was well-tolerated and highly immunogenic in healthy participants [34]. Interim results from the phase 3 trial showed 91.6% efficacy against COVID-19 infection with a good safety profile in a large cohort (21,977 adults: vaccine group ($n = 16,501$), placebo group ($n = 5476$)). The most common adverse events were flu-like illness, injection site reactions, headache, and asthenia. Most of the reported adverse events (7485 (94.0%)) were grade 1; 451 (5.66%) were grade 2, and 30 (0.38%) were grade 3. In total, 68 participants (45 in the vaccine group; 23 in the placebo group) reported serious adverse events. However, none of the serious adverse events were associated with vaccination, which was confirmed by the independent data monitoring committee (IDMC) [34,35].

In the analysis of humoral immune response (titer of neutralizing antibodies against SARS-CoV-2 glycoprotein S), all participants in the vaccine group (on day 21 after dose two) had significantly higher antibody titers with a geometric mean titer (GMT) of 8996 (95% CI 7610–10.635), and a seroconversion rate of 98.25%, compared with the day of administration of the first dose. In the analysis of cellular immune response (IFN-γ secretion upon SARS-CoV-2 glycoprotein S restimulation in culture), all participants in the vaccine group (on day 7 after dose two) had significantly higher levels of IFN-γ (median 32.77 pg/mL (QR 13.94–50.76)), compared with the day of administration of the first dose. In addition to robust cellular and humoral immunity with a good safety profile, the Sputnik V may also induce a broad antibody response with the ability to recognize a wide variety of epitopes on SARS-CoV-2 glycoprotein S, as is suggested by a study conducted by Martynova et al. in 40 individuals vaccinated with the Sputnik V vaccine [44,45].

### 2.2. ChAdOx1-S (Prime) and BNT162b2 (Booster) Regimen

Rare but potentially life-threatening cases of thromboembolic events reported after the administration of the adenovirus vector vaccines ((ChAdOx1 (AstraZeneca/Oxford, UK) and Ad26.COV2.S (Janssen)) have influenced vaccine administration policies in many Western countries [10]. For instance, the British government recommends individuals under 40 years of age should receive a COVID-19 vaccine other than the Janssen vaccine. In addition, countries are now recommending that individuals who have been primed with an adenovirus vector-based vaccine should receive an alternative vaccine as their second booster dose. As a result, many people have received a heterologous prime-boost vaccination with ChAdOx1 as a primer and mRNA vaccines such as the BNT162b2 (Pfizer–

BioNTech) as a booster [22,31,46–49]. Early results from some studies suggest that a ChAdOx1 (prime), BNT162b2 (booster) regimen may induce a robust immunological response with a tolerable reactogenicity profile [10,12,22,50,51].

A phase 2, open-label, randomized, controlled trial study on 676 adults (vaccine group = 450, placebo group = 226) aged 18–60 years reported a robust immunologic response with a safe reactogenicity profile [10,12,22,49,50]. Time elapsed between ChAdOx1-S and BNT162b2 administration varied between 8 to 12 weeks (8 to 9 weeks for 411 (61%) and between 10 to 12 weeks for 263 (39%) participants). They noted that most of the adverse events were mild ($n$ = 1210 (68%)) or moderate ($n$ = 530 (30%)), with injection site pain ($n$ = 395 (88%)), induration ($n$ = 159 (35%)), headache ($n$ = 199 (44%)), and myalgia ($n$ = 194 (43%)) as the most reported adverse events. No serious adverse events were reported [10]. In the analysis of humoral immune response (titer of neutralizing antibodies against SARS-CoV-2 glycoprotein S) in the vaccine group (on day 0 and day 14 after dose two), the GMT of antibody titers increased from 71·46 BAU/mL at baseline (day 0 after dose two) to 7756·68 BAU/mL at day 14 ($p < 0.0001$)). In the analysis of cellular immune response (IFN-$\gamma$ secretion upon SARS-CoV-2 glycoprotein S restimulation in culture), all participants in the vaccine group (on day 14 after dose two) had significantly higher levels of IFN-$\gamma$ (GMT of 521.22 pg/mL) compared with the placebo group (122.67 pg/mL; $p < 0.0001$) [10].

Various studies on non-COVID-19 vaccines have shown that a heterologous prime-boost can be more immunologic than a homologous prime-boost regimen [5,14–16,29,30]. However, such comparative studies are limited to COVID-19 vaccines. In an observational study by Schmidt et al. on 96 adults, the heterologous vaccine regimen induced spike-protein-specific IgG, neutralizing antibodies, and spike-protein-specific CD4+ T cells, the levels of which were significantly higher than after homologous vector vaccine boost ($n$ = 55), and higher or comparable in magnitude to homologous mRNA vaccine regimens ($n$ = 62). In addition, spike-protein-specific CD8+ T cell levels after heterologous vaccination were significantly higher than after both homologous regimens [52]. Similar findings have been seen in larger nationwide cohort studies, which also noted a similar vaccine efficacy (protection) against SARS-CoV-2 infection after heterologous ChAdOx1/mRNA-based COVID-19 vaccine prime-boost (79–88%) compared with after a homologous mRNA-based COVID-19 prime-boost (80–90%) [53,54].

In a UK-based multi-center, randomized trial, all four prime-boost permutations of the ChAdOx1-S and BNT162b2 vaccines were evaluated at 28-day and 84-day prime-boost intervals on 830 participants (age 50 and over). They found that both heterologous schedules (at 28-day and 84-day prime-boost intervals) induced greater systemic reactogenicity following the boost dose than their homologous counterparts, with higher rates of fever, chills, fatigue, headache, joint pain, malaise, and muscle ache [12]. While interpreting these findings, it is also important to note that these data were obtained in participants aged 50 years and older, and reactogenicity in the older age group might be higher than in the younger age group [55,56]. The author also suggested that prophylactic use of paracetamol after immunization may help mitigate these vaccine-related side-effects [12,22,57].

### 2.3. mRNA-1273, Ad26.COV2.S, and BNT162b2

In a US-based multi-center (10 centers) phase 1/2 open-label clinical trial, all nine prime-boost permutations of the three EUA approved by the US Food and Drug Administration (FDA) (Moderna mRNA-1273, Janssen Ad26.COV2.S, and Pfizer-BioNTech BNT162b2) COVID-19 vaccines were evaluated for safety, reactogenicity, and humoral immunogenicity on study days 15 and 29 in 458 adults (154 received mRNA-1273, 150 received Ad26.CoV2.S, and 153 received BNT162b2 booster vaccines) [36,52]. They found (based on non-peer-reviewed published data) that both heterologous regimens increased antibody titers 6.2–76-times compared to a 4.2–20-times increase after a homologous regimen. They also noted a similar reactogenicity profile between the groups with site pain, malaise, headache, and myalgia as the most common side effects noted after booster administration.

### 3. Heterologous Prime-Boost: Where It Might Help

As we continue our fight against COVID-19 with efforts to improve vaccine production worldwide, the growing disparity between wealthy and developing nations regarding access to vaccines has the potential to upend our entire effort. As of November 2021, more than 7.6 billion vaccine doses were administered, with 187.2 million US individuals (65.3% of the population) being fully vaccinated as of November 2021. These numbers are much worse in developing countries. For instance, in low-income countries, only 5% of people have received at least one dose of a COVID-19 vaccine [58]. The use of heterologous booster vaccines with similar reactogenic and immunogenic profiles may help simplify the logistical bottleneck of the COVID-19 vaccine supply chain (fluctuating vaccine supplies). A shortage of vaccine supply might result in delayed administration of the second dose. Therefore, this heterologous prime-boost strategy can help the vulnerable population, especially in low-income countries, get fully vaccinated [52,59].

In addition, there is some promising evidence that a heterologous prime-boost regimen may help immunosuppressed individuals mount a stronger immune response than with a homologous regimen [51]. In a study on 40 transplant recipients, Schmidt et al. reported that heterologous vaccination (ChAdOx1 and mRNA-based) led to higher antibody and CD4 T-cell response compared to a homologous regimen of mRNA-based (mRNA-1273 and BNT162b2) COVID-19 vaccine.

Based on these considerations, in October 2021, the Center of Disease Control and Preventions' (CDC's) Advisory Committee on Immunization Practices (ACIP) issued guidance supporting the "mix-and-match" heterologous boost COVID-19 vaccine strategy [13].

### 4. Limitations

This study has several limitations. Limited number of study participants, and lack of consistency in study selection (selection bias), designs, target population, prime-boost interval, homologous/heterologous immunization regimens, measurement methods, and outcome measures may limit the direct comparisons of immunogenicity and reactogenicity between different studies directly. In addition, most of the data reported were collected from US and European countries; this may limit the generalizability of these findings. Our study only included data from English language articles and, therefore, may have missed relevant information from other language-based sources.

### 5. Conclusions

A heterologous COVID-19 prime-boost vaccine regimen is safe and effective against SARS-CoV-2 infections. Heterologous prime-boost regimens may induce comparable or higher antibody (spike protein) titers than homologous prime-boost. Heterologous prime-boost demonstrated a similar reactogenicity profile to the homologous prime-boost regimen. However, further studies are needed to evaluate the long-term safety and immunogenicity of heterologous prime-boost regimens. Heterologous prime-boost vaccination may help induce a stronger immune response in immunocompromised individuals and improve vaccination drive in countries and regions facing fluctuating supplies of the various vaccines. However, these findings are based on a limited number of preliminary studies. Large scale multicenter prospective, longitudinal studies are needed not only to determine the safety and effectiveness of heterologous COVID-19 prime-boost vaccine regimens, but also to develop standardized heterologous COVID-19 vaccination protocols.

**Author Contributions:** Study concepts and design—R.S., A.B.S. and S.P.; literature research—I.G.; manuscript preparation—I.G.; manuscript editing and final approval—R.S., A.B.S. and S.P. All authors have read and agreed to the published version of the manuscript.

**Funding:** This research received no external funding.

**Institutional Review Board Statement:** Not applicable.

**Informed Consent Statement:** Not applicable.

**Data Availability Statement:** Not applicable.

**Conflicts of Interest:** The authors confirm that this article content has no conflict of interest.

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
