# Peer review of "Mix-and-Match COVID-19 Vaccinations (Heterologous Boost): A Review"

_2036-7449, doi:10.3390/idr14040057_

Round 1
Reviewer 1 Report
Inferring that heterologus prime-boost for COVID-19 vaccine induces a more effective immune response against SARS-CoV2 infection than homologous prime-boost and is safe by introducing various clinical research reports around the world. is doing. This information is extremely useful for the effective use of the COVID-19 vaccine in the future, and may be excellent in inducing a strong immune response, especially for immunocompromised individuals. It is also extremely advantageous in recommending stable vaccination even in the current situation where the vaccine supply situation is fluctuating.
However, the scientific basis for why heterologus prime-boost induces a more effective immune response against SARS-CoV2 infection than homologous prime-boost and is safe is not clear, and it is necessary to clarify it. There is. In particular, what kind of difference in immune induction is observed due to different vaccine dosage types while using the same vaccine antigen, what kind of difference exists in the induction of humoral immunity and cell-mediated immunity, and its immunology It seems that the usefulness of heterologus prime-boost cannot be clearly explained without clarifying some reason.
In this review, the above points are not described much, and it is all about the list of clinical reports. Therefore, the authors should explain what is clear and unclear to date about what I am concerned about. And they should consider how logically it can be explained to the idea that heterologus prime-boost induces a more effective immune response against SARS-CoV2 infection and is safer than homologous prime-boost.
Author Response
Dear reviewer, thank you for the astute observation. We agree that it is important to understand the underlying mechanism for why heterologous prime-boost may induce a more effective immune response than homologous prime-boost vaccination. However, the mechanism and reason heterologous prime-boost are more immunogenic than the homologous prime-boost regimen remains poorly understood and needs further studies. We have included limited information available on the proposed mechanism of heterologous prime-boost. “Although the exact mechanism behind this is not well understood [14-16, 29, 30]. Some studies have suggested that priming with DNA based vaccine may improve the avidity of antibody response to a protein-based vaccine. It is hypothesized that in vivo production of antigen by DNA or mRNA-based vaccine may lead to the production of memory B cells specific for the conformational domains of the antigen. Supportive evidence was reported by Vaine et al. in a rabbit model. They demonstrated the presence of conformational sensitive antibodies following priming by a DNA-based vaccine (for in vivo delivery of HIV-1 gp120 antigen).”

Reviewer 2 Report
Could you define how articles have been retrieved and selected? The methods section is missing. please add.
Add in the title that it is a review article.
When speaking about seroconversion, please specify the threshold set.
For each trial, please report the id.
In table 1 please add if the study received funds and the conflict of interest declared by the authors. Please, also add information regarding participants' characteristics (at least sex, age, and health condition)
Results and discussion are mixed up. Please, consider revising the manuscript clearly dividing the results from the discussion.
Did the authors register the protocol of the current review? If yes, please add. If not, please mention it in the limitations.
In the conclusions, please highlight the preliminary nature of your results and the difficulties in drawing conclusive recommendations because of the limitations.
In the limitations, please also mention the low number of total studies included and the arbitrary method used to select the manuscripts (selection bias).
Author Response
- Could you define how articles have been retrieved and selected? The methods section is missing. Please add.-
Response: Dear reviewer, thank you for your comment. We intend this article to be a narrative review and is structured as such. Therefore a literature search methodology is not included in the article. We have made all the effort to include relevant and pivotal studies in the article.
- Add in the title that it is a review article. –
Response: Thank you for the suggestion. We have modified the title of the article.
- When speaking about seroconversion, please specify the threshold set.
Response: Thank you for your comment. We have added this information
- For each trial, please report the id.
Response: Thank you for your comment. We have added this information
- In table 1, please add if the study received funds and the conflict of interest declared by the authors. Please, also add information regarding participants' characteristics (at least sex, age, and health condition)
Response: Thank you for your comment. We have added this information in the table.
- Results and discussion are mixed up. Please, consider revising the manuscript clearly dividing the results from the discussion.
Response: Thank you for your comment. As this is a narrative review, we have not divided the manuscript into result and discussion sections. However, we have failed to make that sufficiently clear. We have amended our manuscript accordingly and now write “Based on the above considerations, the purpose of this article is to provide an in-depth review of the heterologous prime-boost COVID-19 vaccine regimen. The immunogenicity and reactogenicity of heterologous prime-boost COVID-19 vaccine regimen, compared to the existing homologous regimen is discussed.”
- Did the authors register the protocol of the current review? If yes, please add. If not, please mention it in the limitations. _
Response: Thank you for your comment. We have not registered the protocol of the current review.
- In the conclusions, please highlight the preliminary nature of your results and the difficulties in drawing conclusive recommendations because of the limitations.
Response: Thank you for your comment. We have added this information to the conclusions section.” However, these findings are based on limited number of preliminary studies. Large scale multicenter prospective, longitudinal studies are needed not only to determine the safety and effectiveness of heterologous COVID-19 prime-boost vaccine regimen but also to develop standardized heterologous COVID-19 vaccination protocols.”
- In the limitations, please also mention the low number of total studies included and the arbitrary method used to select the manuscripts (selection bias). –
Response: Thank you for your comment. We have added this information to the limitations section. “Limited number of study participants, lack of consistency in study selection (selection bias), designs, target population, prime-boost interval, homologous/heterologous immunization regimens, measurement methods, and outcome measures may limit the direct comparisons of immunogenicity and reactogenicity between different studies directly”.

Round 2
Reviewer 1 Report
In the revised manuscript, the text is appropriately revised according to my points. I consider this edition to be acceptable to this magazine.
Reviewer 2 Report
I am satisfied with the changes made by the authors